

# Estimating ankle joint angle from skeletal geometry: a mechanical model of the calcaneal lever in terrestrial mammals

Fumihiro Mizuno[1] and Shin-ichi Fujiwara[2]

[1] National Museum of Nature and Science, Tsukuba, Ibaraki, Japan
[2] Nagoya University Museum, Nagoya University, Nagoya, Aichi, Japan

## ABSTRACT

**Background:** The ankle joint angle, typically measured between the tibia and metatarsus, shows only a small range of movement during the stance phase and remains relatively constant within species, but varies across taxa. This variation influences traits such as stride length, posture, and locomotor function. While joint angles are readily observable in living animals, they cannot be directly measured in extinct species, for which only skeletal remains are available. Therefore, estimating ankle joint posture from skeletal geometry is important for reconstructing locomotion in both extant and extinct mammals. In this study, we propose a mechanical model of the ankle extensor apparatus to estimate ankle joint angle from bones and test whether the muscle-lever system aligns consistently with skeletal features across taxa.

**Methods:** We developed a simplified mechanical model of the ankle extensor apparatus to calculate ankle extensor moment arm defined as the perpendicular distance from the ankle joint center to the muscle force line of action, which was assumed to be parallel to the tibia. To verify the Achilles tendon runs parallel to the tibia across taxa, dissections were performed on cadavers of 24 species in seven orders. We compared observed angle ($\theta_{obs}$) from 26 species of zoo-kept terrestrial mammals, covering various body mass and locomotor modes, with estimated angle ($\theta_{est}$) from skeletal specimens of the same species. $\theta_{obs}$ was the mean tibia–metatarsus angle during the stance phase, recorded laterally with a high-speed camera. $\theta_{est}$ was measured on reassembled skeletal specimens as the ankle joint angle that maximized the extensor moment arm in the model. Phylogenetic comparative methods, including phylogenetic ANOVA and PGLS, were applied to analyze relationships among $\theta_{obs}$, $\theta_{est}$, body mass, and locomotor mode based on a time-calibrated phylogeny.

**Results:** Dissections confirmed the Achilles tendon runs nearly parallel to the tibia across species. Stance phase ankle joint rotations were small. Therefore, $\theta_{obs}$ could be considered as representative for each species. Over 85% of the studied species maintained their ankle joint angle at which the mechanical advantage of the calcaneal lever was greater than 0.9. No significant differences in the mechanical advantage of the calcaneal lever were found among locomotor modes or taxonomic orders. A strong positive correlation was observed between $\theta_{obs}$ and $\theta_{est}$ ($\rho = 0.70$, $p < 0.001$).

**Conclusion:** Our mechanical model could estimate $\theta_{est}$ from skeletal morphology that closely match $\theta_{obs}$ during stance phase. Despite interspecific variation of $\theta_{obs}$, the mechanical advantage of the calcaneal lever remains within a narrow range,

Corresponding author
Fumihiro Mizuno,
fmizuno86@kahaku.go.jp

suggesting mechanical optimization of the ankle extensor apparatus across terrestrial mammals. This model informs postural reconstruction in extinct species.

# INTRODUCTION

The hindlimbs perform some essential functions for terrestrial life, such as supporting body mass, generating propulsion, and lifting the body. During walking, in particular, the hindlimbs contribute to forming an inverted pendulum (*Cavagna, Heglund & Taylor, 1977*; *Alexander & Jayes, 1978a*, *1978b*; *Hildebrand, 1984*; *Hildebrand & Hurley, 1985*; *Alexander, 1991*; *Griffin, Main & Farley, 2004*) or spring-loaded inverted pendulum (SLIP) model (*Geyer, Seyfarth & Blickhan, 2006*), both of which facilitate efficient locomotion by generating propulsive energy. The SLIP model proposes that the limbs act as springs, while also emphasizing the importance of optimized stiffness (*Blickhan & Full, 1993*; *Geyer, Seyfarth & Blickhan, 2006*). In these models, the rod of the inverted pendulum is defined as the line between the grounding point of the metatarsus and the center of mass (*Cavagna, Heglund & Taylor, 1977*; *Alexander & Jayes, 1978a*, *1978b*; *Blickhan & Full, 1993*; *Griffin, Main & Farley, 2004*; *Geyer, Seyfarth & Blickhan, 2006*); therefore, the rod length is governed by the joint angles, particularly those of the knee joint and the ankle joint in the hindlimbs. The previous studies showed that the knee joint angles among mammalian species do not change drastically—remain relatively constant with small changes—during the stance phase, and differ between species (*Manter, 1938*; *Gray, 1944*; *Goslow, Reinking & Stuart, 1973*; *Goslow et al., 1981*; *Alexander & Jayes, 1983*; *Inuzuka, 1996*; *Fischer et al., 2002*; *McGowan, Baudinette & Biewener, 2005*; *Dutto et al., 2006*; *Day & Jayne, 2007*; *Ren et al., 2008*; *Patel et al., 2013*). The variations in the joint angles contribute to differences in body height, locomotion, and biology between species. However, *Mizuno & Kohno (2023)* showed that the angle between the line of the action of the *musculus semimembranosus* and the tibia resembles across mammals, even when these taxa belong to different orders. Similarly, *Fujiwara (2009)* and *Fujiwara & Hutchinson (2012)* showed that the angle between the olecranon process and the *m. triceps brachii* remains similar despite differences in the elbow joint angle typically measured between the humerus and ulna. These studies indicate that while joint angles observable from outside the body vary across species, may be conserved across mammals due to skeletal morphology. Therefore, identifying the mechanisms that determine such skeletal angles enables the reconstruction of limb postures in extinct animals, whose joint angles cannot be directly observed.

The foot is a part of the inverted pendulum of the entire hindlimb, and also functions as a lever system driven by extensor muscles to generate greater ground reaction force (*Gregory, 1912*; *Manter, 1938*; *Biewener, 1989*, *1990*; *McGowan, Baudinette & Biewener, 2005*; *Deforth et al., 2019*). The upward rotation of the calcaneal tuber, caused by contraction of these muscles, induces the downward rotation of the metatarsus, thereby generating ground reaction force. Anatomically, the extensor muscles—specifically the

*m. gastrocnemius*, *m. soleus*, and *m. flexor digitorum superficialis*—contribute to raising the calcaneal tuber. These extensor muscles form the *tendo calcaneus* (Achilles tendon), although the *m. soleus* is absent or weakened in some taxa. This muscle originates from the flexor side of the knee joint—specifically, the distal femur and the proximal tibia and fibula—and inserts into the distal end of the calcaneal tuber *via* the Achilles tendon, running nearly parallel to the tibia (*Kato & Yamauchi, 2003*; *König, Liebich & Bragulla, 2007*). Therefore, the net force vector produced by the extensor muscles can be assumed to run parallel to the axis of the tibia. The functional role of the ankle extensors during the stance phase to support body mass and generate propulsive force against gravity has been well established (*Engberg & Lundberg, 1969*; *Goslow, Reinking & Stuart, 1973*; *Rasmussen, Chan & Goslow, 1978*; *Walmsley, Hodgson & Burke, 1978*; *Goslow et al., 1981*; *Nicolopoulos-Stournaras & Iles, 1984*; *Whelan, Hiebert & Pearson, 1995*). In terrestrial mammals, particularly in unguligrade and digitigrade species, the ankle joint is actively extended to counteract gravitational flexion forces during both standing and stance phase. Accordingly, an extensor torque must be generated at the ankle joint, with the calcaneal tuber playing a central role in its propagation. This extensor torque arises from the force applied *via* the calcaneal tuber and the internal moment arm, defined as the perpendicular distance between the joint's center of rotation and the line of action of the muscle force (*Fujiwara, 2009*). Given this biomechanical context, it is reasonable to hypothesize that mammals adopt joint angles that maximize this moment arm during the stance phase, thereby enhancing mechanical advantage and stability.

We hypothesize that (i) the functional ankle extension angle (FAEA), defined as the angle between the line of action of the extensor muscles and calcaneal tuber, is maintained within a range that optimizes muscle force production and mechanical advantage during the stance phase of walking in terrestrial mammals; (ii) FAEA is similar among species with different observable ankle joint angle ($\theta$) which is typically measured; and (iii) $\theta$ can be estimated from a mechanical model of the ankle extensor apparatus. This study aims to test these hypotheses using a mechanical model based on the skeletal geometry of each target species, and to provide insight into the fundamental constraints shaping limb posture, which may aid in the functional interpretation of extinct taxa.

## MATERIALS AND METHODS

### Mechanical model

To examine the hypotheses, we constructed a simplified mechanical model of joint rotation. For instance, the trochlea of the talus was approximated as a circular arc, and the articulations between the foot bones were assumed to be rigid. This simplification was necessary to allow general comparison across species, as skeletal morphology varies not only between species but also among individuals. Assuming the foot as a simple body of rotation about the ankle joint (Fig. 1) during extension and flexion, the ankle extensor torque ($\tau$) is generated by the extensor muscles, which inserts at the most distal point on the calcaneus (C) (Fig. 1). $\tau$ can be expressed by the following equation:

$$\tau = l \times F \tag{1}$$

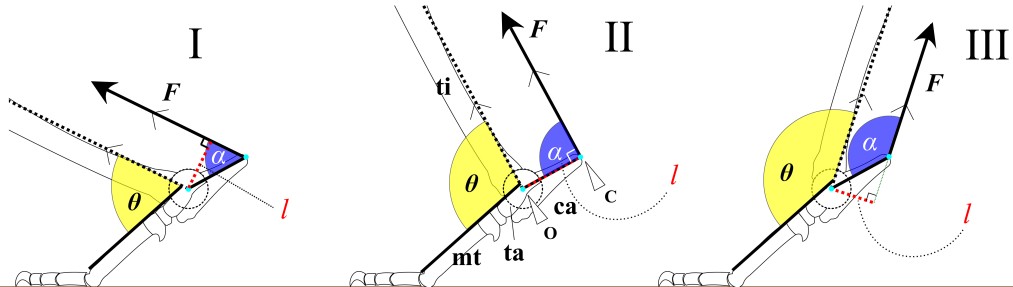

**Figure 1 Mechanical models of the ankle extensor apparatus.** Three models show (I) $a < 90°$, (II) $a = 90°$, and (III) $a > 90°$. Each yellow arc represents the ankle joint angle ($\theta$) between the tibia and foot. Each blue arc represents the angle ($a$) between the line of action of the extensor muscle ($F$) and the line connecting the center of the ankle joint rotation (O) to the most distal point on the calcaneus (C). Each red dashed line represents the moment arm ($l$). $\theta$ at position (II) is defined as $\theta_{est}$. Abbreviations: ca, calcaneus; mt, metatarsus III; and ta, talus.

where $F$ is a contractile force of the extensor muscles, and $l$ is the moment arm length. The moment arm $l$ is defined as the perpendicular distance from the center of the ankle joint rotation (O) to the line of action of $F$ (Fig. 1). If $F$ is constant, $\tau$ is directly proportional to $l$. The moment arm $l$ can be calculated using the following equation:

$$l = OC \cdot \sin \alpha \tag{2}$$

where OC is the distance between O and C, and $\alpha$ represents the value of FAEA. Since the length OC is fixed for a given specimen, the value of $l$ depends entirely on the angle $\alpha$ (Fig. 1). The function $\sin \alpha$ reaches its maximum when $\alpha$ is 90° (Fig. 1). Therefore, $l$ is maximized ($\sin \alpha = 1$) when the line OC is perpendicular to the tibia and vector $F$. The value of $\sin \alpha$ can thus be regarded as the mechanical advantage of the calcaneal lever.

Assuming that the $F$ runs parallel to the shaft of the tibia and that the articulations between the tarsal and metatarsal bones are rigid, we defined the estimated ankle joint angle ($\theta_{est}$) as the angle at which $\alpha = 90°$ (Fig. 1). At this angle, the mechanical advantage of the calcaneal lever ($\sin \alpha$) is maximized. The mechanical advantage of the calcaneal lever decreases when the ankle joint angle $\theta$ deviates above or below this value $\theta_{est}$. To investigate the directional relationship between the tibia and the Achilles tendon, anatomical observations were performed on several cadaveric specimens. Details of these observations are provided below. All angles (e.g., $\theta$ and $\alpha$) are expressed in degrees unless otherwise noted. $\theta$ varies depending on the specimen (Fig. 2).

## Dissection

We dissected cadavers donated to the National Museum of Nature and Science (NMNS) by zoos to confirm that the Achilles tendon runs parallel to the tibia, and that the orientation is consistent across different species. The species dissected belong to seven orders, 23 genera, and 24 species (Table 1). All individuals were adults. We followed the ethical guidelines of the NMNS during the dissection.

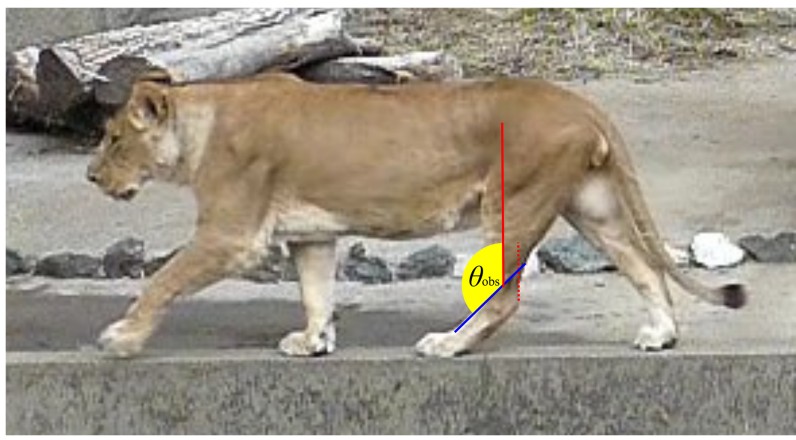

**Figure 2 An example image used to collect ankle joint angle data from living individuals.** The red dashed line represents the Achilles tendon, while the red solid line indicates the tibial shaft, which runs parallel to the Achilles tendon. The blue solid line shows the dorsal margin of the third metatarsal (metatarsus III). The yellow fan illustrates the ankle joint angle ($\theta_{obs}$) being measured.

**Table 1 A list of dissected species by first author.**

| Species | Location |
|---|---|
| **Artiodactyla** | |
| *Bos taurus* | NMNS |
| *Capricornis crispus* | NMNS |
| *Cervus nippon yakusimae* | NMNS |
| *Elaphurus davidianus* | NMNS |
| *Giraffa camelopardalis* | NMNS |
| *Rangifer tarandus* | NMNS |
| **Perissodactyla** | |
| *Equus caballus* (pony) | NMNS |
| *Equus zebra* | NMNS |
| *Tapirus terrestris* | KPM |
| **Carnivora** | |
| *Aonyx cinerea* | NMNS |
| *Acinonyx jubatus* | NMNS |
| *Canis lupus familialis* | NMNS |
| *Chrysocyon brachyurus* | NMNS |
| *Panthera leo* | NMNS |
| *Suricata suricatta* | NMNS |
| *Vulpes vulpes* | NU |
| **Primates** | |
| *Cercopithecus neglectus* | NMNS |
| *Erythrocebus patas* | NMNS |
| *Macaca fuscata* | NMNS |

(Continued)

| Table 1 (continued) | |
|---|---|
| **Species** | **Location** |
| **Rodentia** | |
| *Castor canadensis* | NMNS |
| *Dolichotis patagonum* | NMNS |
| *Galea musteloides* | NMNS |
| **Diprotodontia** | |
| *Petrogale xanthopus* | NMNS |

**Note:**
Abbreviations: NMNS, National Museum of Nature and Science, Tsukuba, Japan; KPM, Kanagawa Prefectural Museum of Natural History, Odawara, Japan; NU is Japan University, Nagoya, Japan.

## Collection from living individuals

We defined the ankle joint ($\theta_{obs}$) as the angle between the tibia and the metatarsus III of living individuals. Data were collected from 26 extant species, including two domestic breeds (Kiso Horse and thoroughbred) within *Equus caballus*. Thus, the total number of species was 26, representing 25 genera and 16 families across nine orders (Table 2). These species were selected to cover a wide range of taxa, body masses (*i.e.*, from 0.06 kg for the Mongolian gerbil to 3,400 kg for white rhinoceros), and locomotor mode (unguligrade, digitigrade, plantigrade, and pentapedal) (Table 2). Pentapedal locomotion refers to the mode used by kangaroos (*O'Connor et al., 2014*). Most target animals are kept in zoos at Higashi Park Zoological Gardens (Aichi, Japan), Higashiyama Zoo and Botanical Garden (Aichi, Japan), Hitachi Kamine Zoo (Ibaraki, Japan), Toyohashi Zoo and Botanical Park (Aichi, Japan), and Ueno Zoological Gardens (Tokyo, Japan), and all observations were conducted with official permissions from these facilities. Mongolian gerbils and a cat were kept by a colleague of the authors, and the thoroughbred was hosed at the equestrian arts club of Tsukuba University Equestrian Team (University of Tsukuba). No individuals showed significant pathologies or malformations. Furthermore, we avoided using juvenile individuals.

High-speed video recordings (420 fps) were obtained using a Casio EX-FH20 digital camera (Japan) mounted on a tripod positioned along public viewing paths. The distance between the camera and the targets varied depending on the exhibition/cage layout. All videos were taken from the lateral side and at nearly the same level as each target animal when they walked voluntarily in a straight line without stopping, turning, or changing speed, and perpendicular to the camera on flat ground. No external stimuli (handlers, markers, and confinement) were applied, and recordings were made passively. Kangaroos were recorded while pentapedal locomotion, which use four legs and a tail, rather than hopping. Some data were cited from previous studies (Table 2).

We defined one stance phase, from touch-down to take-off, as one step during stance phase. Each step was screen-captured from the videos using GOM player (Gretech Corporation, Seoul, South Korea). The captured images of each step were divided into five equal time intervals to yield six images per stance phase. On each of the six images, two lines were drawn using Inkscape (ver. 1.2, Inkscape Project), and the angle between them

**Table 2 List of the studied species, with the number of the steps used for the analysis, the locations where the data of the steps were collected, number of the skeletal specimens used for the measurements, and the list of the specimens.**

| Order, Family | Species (common name) | Steps [n] | Locality or reference | Skeletons [n] | Specimen |
|---|---|---|---|---|---|
| **Artiodactyla** | | | | | |
| **Bovidae** | | | | | |
| | *Ammotragus lervia* (Barbary sheep) | 2 | UZ | 1 | NMNS M42950 |
| | *Capra hircus* (goat) | 4 | Co1 | 1 | KPM NF-1004771 |
| | *Capricornis crispus* (Japanese serow) | 2 | HZBG | 3 | NMNS M34472; NMNS M35895; TMNH MA304 |
| **Cervidae** | | | | | |
| | *Cervus nippon yesoensis* (Yezo shika deer) | 2 | UZ | 1 | NMNS M39257 |
| | *Rangifer tarandus* (reindeer) | 2 | HZBG | 2 | NMNS M31343; NMNS M34433 |
| **Giraffidae** | | | | | |
| | *Giraffa reticulata* (reticulated giraffe) | 9 | UZ | 3 | NMNS M33232; NMNS M34445; NMNS M47162 |
| **Hippopotamidae** | | | | | |
| | *Choeropsis liberiensis* (pygmy hippopotamus) | 3 | UZ | 1 | NMNS M36955 |
| **Perissodactyla** | | | | | |
| **Equidae** | | | | | |
| | *Equus caballus* (Kiso horse) | 3 | TZBP | 1 | NMNS M31212 |
| | *Equus caballus* (thoroughbred) | 3 | TET | 2 | NMNS M75179; NMNS M77948 |
| **Rhinocerotidae** | | | | | |
| | *Ceratotherium simum* (white rhinoceros) | 2 | TZBP; *Day & Jayne (2007)* | 3 | NMNS M31319; NMNS M31455; NMNS M37727 |
| | *Rhinoceros unicornis* (Indian rhinoceros) | 5 | HZBG | 1 | KPM NF1002747 |
| **Tapiridae** | | | | | |
| | *Tapirus terrestris* (South American tapir) | 2 | HZBG | 2 | NMNS M31505; OMNH unnumbered |
| **Carnivora** | | | | | |
| **Canidae** | | | | | |
| | *Canis lupus* (wolf) | 4 | HZBG | 2 | NMNS M43287; OMNH M440 |
| | *Chrysocyon brachyurus* (maned wolf) | 3 | UZ | 2 | NMNS M36655; NMNS M53024 |
| **Felidae** | | | | | |
| | *Acinonyx jubatus* (cheetah) | (125) | *Day & Jayne (2007)* | 2 | NMNS M31465: NMNS M37279 |

(Continued)

| Order, Family | Species (common name) | Steps [n] | Locality or reference | Skeletons [n] | Specimen |
|---|---|---|---|---|---|
| | *Felis silvestris catus* (cat) | 3 | Co1 | 3 | NMNS M36028; NMNS M36030; NMNS M76763 |
| | *Panthera leo* (lion) | 6 (119) | TZBP; *Day & Jayne (2007)* | 3 | NMNS M37728; NMNS M42648; NMNS M43347 |
| | *Panthera tigris* (Sumatran tiger) | 3 (125) | HZBG; *Day & Jayne (2007)* | 4 | NMNS M38483; NMNS M38486; OMNH M2401; TMNH MA318 |
| **Herpestidae** | | | | | |
| | *Suricata suricatta* (meerkat) | 3 | HZBG | 2 | NMNS 33399; NMNS M54629 |
| **Primates** | | | | | |
| **Cercopithecidae** | | | | | |
| | *Cercopithecus neglectus* (De Brazza's monkey) | 2 | UZ | 2 | NMNS M71712; NMNS M75202 |
| | *Macaca fuscata* (Japanese macaque) | 3 | KZ | 3 | NMNS M36028; NMNS M36074; NMNS M63048 |
| **Rodentia** | | | | | |
| **Caviidae** | | | | | |
| | *Dolichotis patagonum* (Patagonian mara) | 2 | HPZG | 3 | NMNS M35595; NMNS M35760; NMNS M35880 |
| | *Galea musteloides* (yellow-toothed cavy) | (73) | *Fischer et al. (2002)* | 1 | NMNS M36015 |
| **Muridae** | | | | | |
| | *Meriones unguiculatus* (Mongolian gerbil) | 2 | Co2 | 1 | NMNS M11702 |
| **Scandentia** | | | | | |
| **Tupaiidae** | | | | | |
| | *Tupaia glis* (common tree shrew) | (84) | *Fischer et al. (2002)* | 1 | NMNS M28531 |
| **Diprotodontia** | | | | | |
| **Macropodidae** | | | | | |
| | *Macropus fuliginosus* (western grey kangaroo) | 1 | HZBG | 1 | NMNS M43349 |
| | *Macropus giganteus* (eastern grey kangaroo) | 3 | UZ | 1 | NMNS M43210 |

**Note:**
Abbreviations for the locality: Co1 and 2, colleagues who keep the animal; HZBG, Higashiyama Zoo and Botanical Garden, Nagoya, Japan; KZ, Hitachi Kamine Zoo, Hitachi, Japan; TET, Tsukuba University Equestrian Team, Tsukuba, Japan; TZBP, Toyohashi Zoo and Botanical Park, Toyohashi, Japan; UZ, Ueno Zoological Garden, Tokyo, Japan.
Abbreviations for the museums: NMNS, National Museum of Nature and Science, Tsukuba, Japan; KPM, Kanagawa Prefectural Museum of Natural History, Odawara, Japan; OMNH, Osaka Museum of Natural History, Osaka, Japan.

was measured: one line extended from the ankle joint and the proximal end of the tibia, parallel to the Achilles tendon, and the other line along the dorsal margin of the metatarsus III (Fig. 2). We used three or more steps for 13 species, two steps for nine species, and one step for one species, from videos. In addition, five species were cited from previous studies including three species for which both our video data and previous studies were used (Table 2). The mean of the measured angles was defined as $\theta_{obs}$.

## Collection from skeletal specimens

The ankle joint angles at which the extensor moment arm is maximized were measured using skeletal specimens. The data were collected for the same 26 species used in the observations of $\theta_{obs}$ (Table 2). Disarticulated hindlimb bones stored at the National Museum of Nature and Science (NMNS: Tsukuba, Japan), Toyohashi Museum of Natural History (Toyohashi, Japan), Kanagawa Prefectural Museum of Natural History (Odawara, Japan), and Osaka Museum of Natural History (Osaka, Japan) (Table 2) were used. A total of 42 skeletal specimens were measured. No specimens showed significant pathologies or malformations. Furthermore, we avoided using juvenile specimens.

We reassembled the metatarsals and tarsal bones (calcaneus, talus, cuneiforms, navicular, and cuboid) using a soft, reusable adhesive (Hittsukimushi, Kokuyo, Osaka, Japan), and then the flexibility of the joint connections was confirmed to be negligibly small. The Mongolian gerbil foot skeleton was stored with the bones connected by tendons and ligaments. The reassembled bones were photographed in lateral view using a digital camera (D3400; Nikon, Tokyo, Japan).

Two lines were then drawn on each image, and the angle between them was measured following procedure described below. First, a circle centered at O, which best fits the ridge of the trochlea of the talus, was drawn. Next, a perpendicular line (line A) to the line segment OC, originating from point O, was drawn. Following this, a line connecting the dorsal tips of the proximal and distal epiphyses on the metatarsal III (line B) was drawn (see description in Mechanical Model section). Finally, the angle between lines A and B was defined as the ankle joint angle where the extensor moment arm maximized for the specimen (Fig. 3). These figures were also created using Inkscape. The average of the angles for each species was defined as $\theta_{est}$.

## Statistical analysis

For statistical analysis, body mass (in grams) was transformed using natural logarithmic scale (ln). The locomotor mode was classified into four groups: plantigrade, digitigrade, unguligrade, and pentapedal (Table 2). Phylogenetic Comparative Methods (PCMs) were applied to all measurements and derived variables. Pairwise comparisons were conducted following the phylogenetic ANOVA (phyANOVA) using phylogenetic t-tests with Holm's correction for multiple testing to determine which pairs had significant differences (*Garland et al., 1993*; *Revell, 2012*). Additionally, the correlation coefficient between $\theta_{obs}$ and $\theta_{est}$ was calculated.

Phylogenetic generalized least squares (PGLS) analyses were performed to examine the following relationships: (1) mechanical advantage of the calcaneal lever and locomotor

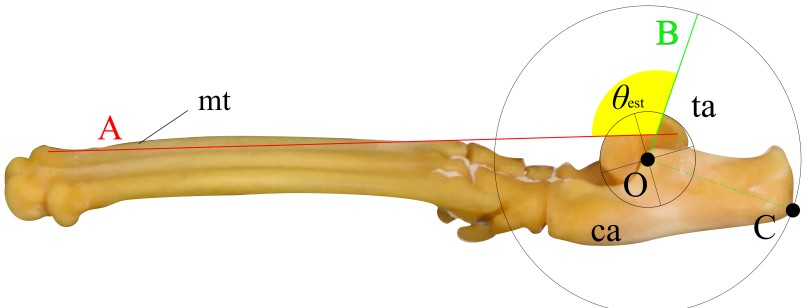

**Figure 3 An example of the picture for measuring the ankle joint angle data from the skeletal specimens using a cat (M36028).** The foot element in connection (tarsal and metatarsal bones) is photographed from the medial view. Point O represents the center of a circle that best fits the edge of the trochlea of the talus. Point C indicates the contact point on the calcaneus when the concentric circle reaches its maximum size. Line A connects the dorsal edges of the proximal and distal epiphyses of metatarsal III. Line B is drawn perpendicular to the line segment OC, originating from point O. The yellow fan represents the measured ankle joint angle ($\theta_{est}$). Abbreviations: ca, calcaneus; mt, metatarsus III; and ta, talus. The scale bar is 10 mm. This foot was rearticurated and taken by the author at NMNS.

mode; (2) mechanical advantage of the calcaneal lever and body mass; and (3) body mass, $\theta_{obs}$ and $\theta_{est}$. The phylogenetic tree used for these analyses was constructed using Timetree: The Timescale of Life (https://timetree.org/).

All statistical analyses were performed using the R software package (ver. 4.4.1, *R Core Team, 2024*).

## RESULTS

The directions of the Achilles tendon and the tibia were nearly parallel in the species dissected in this study (Fig. 4). This parallel orientation of the Achilles tendon relative to the tibia was consistently observed over a wide range of joint angles. Therefore, the direction of the line of action of the ankle joint extensor muscle can be approximated by the tibia. Additionally, the transitions of the ankle joint angle during the stance phase were checked to ensure the $\theta_{obs}$ represented the angle of the target species. According to the line graph of the ankle joint angle transitions, these were gradual and the maximum differences between the average angle ($\theta_{obs}$) and the maximum or minimum angle were small in the many studied species (Fig. 5): in 22 species of the 27 studied species, the maximum differences were less than 20 degrees. Thus, $\theta_{obs}$ was considered representative for each species and was suitable for comparison with $\theta_{est}$. The Mongolian gerbil which had the smallest body mass, 0.06 kg, had both the minimum $\theta_{obs}$ and $\theta_{est}$ (64° and 95°, respectively). The white rhinoceros, which has the largest body mass, 3,400 kg, had the second maximum $\theta_{est}$ (141°). Thoroughbred, with the fourth largest body mass (518 kg), had the maximum $\theta_{obs}$ (157°), whereas the maximum $\theta_{est}$ was 146° for the Indian rhinoceros, which had the second largest body mass (2,100 kg) (Table 3). The Japanese serow, reindeer, and Japanese macaque had the smallest absolute difference between $\theta_{est}$ and $\theta_{obs}$ (0°), and the thoroughbred had the largest difference, 38° (Table 3).

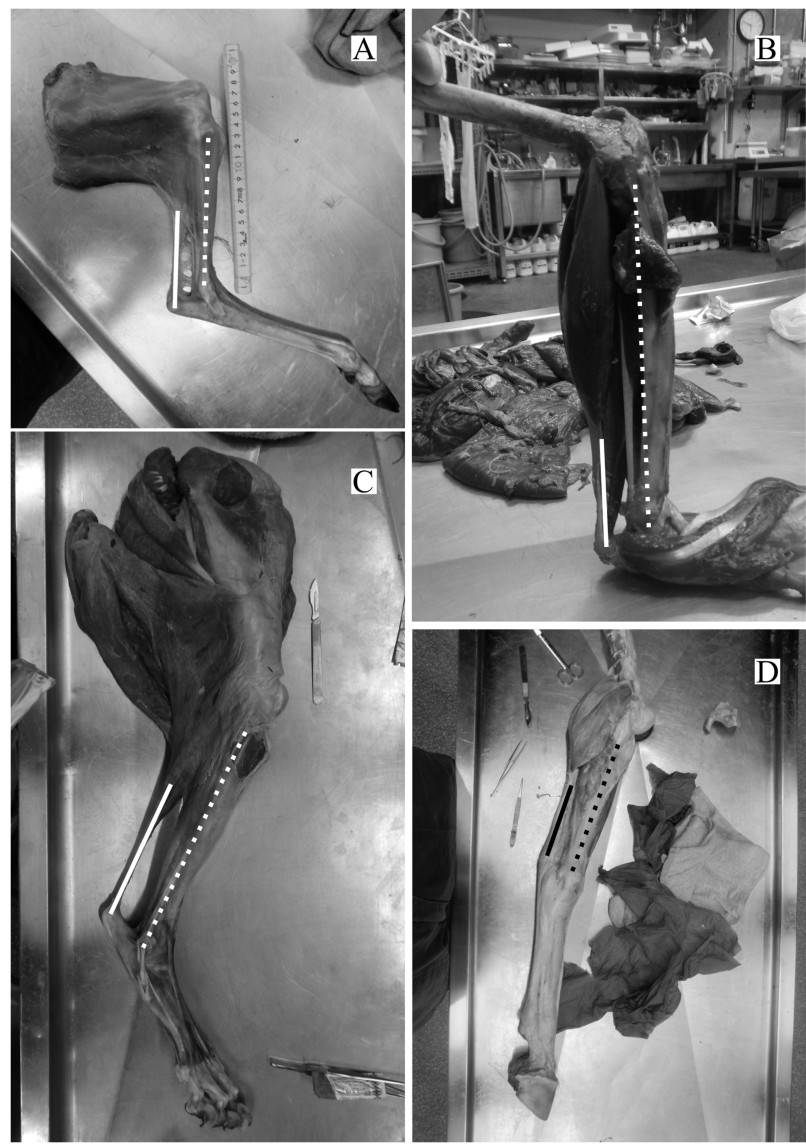

**Figure 4 Some examples of the hind limbs dissected by first author.** (A) *Cervus nippon yakusimae*; (B) *Macaca fuscata*; (C) *Panthera leo*; (D) *Equus caballus* (pony). Solid lines show the Achilles tendon, and broken lines showed the shaft of the tibia. Note that these lines run nearly parallel with each other. The author performed the dissections and took these photos at NMNS.

In 24 species (including the Kiso horse), the value of the mechanical advantage of the calcaneal lever exceeded 0.9; among these, 18 species had values greater than 0.95. Three species had the mechanical advantage of the calcaneal lever between 0.8 and 0.9. The thoroughbred had the lowest mechanical advantage of the calcaneal lever at 0.79 (Table 3 and Fig. 6). PhyANOVA and pairwise comparisons were conducted to assess significant differences in the mechanical advantage of the calcaneal lever among locomotor modes. The phyANOVA showed no significant differences ($p = 0.602$), and all pairwise Holm-adjusted $p$-values were equal to 1.00; therefore no significant differences in the

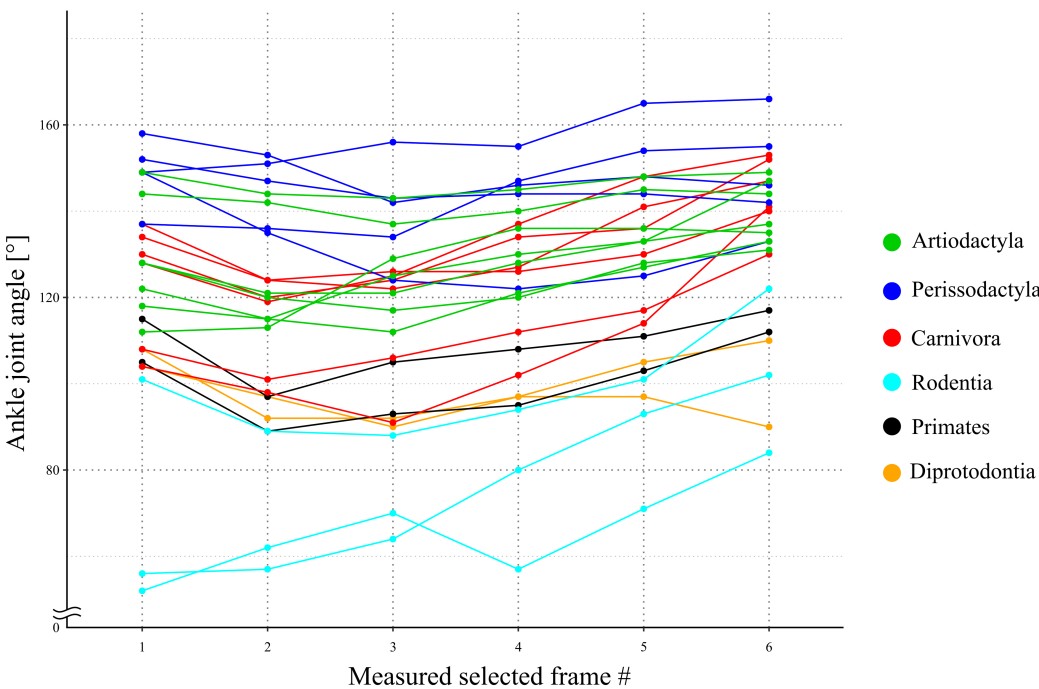

**Figure 5 A line graph of the angle transition of target animals.** X-axis shows the number of the measured selected frames between the beginning and the end of the stance phase. Y-axis shows the ankle joint angle. The averages of these angles were defined as $\theta_{obs}$ [°] for each species. Each color shows the taxonomic group (order) of the target animals and the representation of each color in the legend.

**Table 3 The values of the observed ankle joint angle ($\theta_{obs}$), estimated ankle joint angle ($\theta_{est}$), efficiency of the calcaneal lever of the ankle joint extensor (sin $\alpha$), body mass, and locomotor mode of the studied species.**

| Order | Family | Species | $\theta_{obs}$ (°) | $\theta_{est}$ (°) | sin $\alpha$ | Body mass (kg) | Locomotor mode |
|---|---|---|---|---|---|---|---|
| Artiodactyla | Bovidae | *Ammotragus lervia* | 142 | 130 | 0.98 | 64 | U |
| | | *Capra hircus* | 130 | 137 | 0.99 | 20 | U |
| | | Japanese serow | 127 | 127 | 1.00 | 38 | U |
| | Cervidae | *Capricornis priscus* | 126 | 123 | 0.99 | 75 | U |
| | | *Rangifer tarandus* | 122 | 121 | 1 | 120 | U |
| | Giraffidae | *Giraffa reticulata* | 146 | 140 | 0.99 | 1,000 | U |
| | Hippopotamidae | *Choeropsis liberiensis* | 124 | 117 | 0.99 | 223 | U |
| Perissodactyla | Equidae | *Equus caballus*: Kiso horse | 144 | 125 | 0.95 | 400 | U |
| | | *Equus caballus*: thoroughbred | 157 | 119 | 0.79 | 518 | U |
| | Rhinocerotidae | *Ceratotherium simum* | 149 | 141 | 0.99 | 3,400 | U |
| | | *Rhinoceros unicornis* | 145 | 146 | 1.00 | 2,100 | U |
| | Tapiridae | *Tapirus terrestris* | 131 | 102 | 0.87 | 200 | U |
| Carnivora | Canidae | *Canis lupus* | 130 | 119 | 0.98 | 40 | D |
| | | *Chrysocyon brachyurus* | 133 | 124 | 0.98 | 24 | D |

| Table 3 (continued) | | | | | | | |
|---|---|---|---|---|---|---|---|
| Order | Family | Species | $\theta_{obs}$ (°) | $\theta_{est}$ (°) | sin $\alpha$ | Body mass (kg) | Locomotor mode |
| | Felidae | *Acinonyx jubatus* | 125 | 112 | 0.97 | 48 | D |
| | | *Felis silvestris catus* | 112 | 109 | 0.99 | 4.8 | D |
| | | *Panthera tigris* | 125 | 114 | 0.98 | 125 | D |
| | | *Panthera leo* | 120 | 108 | 0.98 | 188 | D |
| | Herpestidae | *Suricata suricatta* | 108 | 121 | 0.98 | 0.7 | D |
| Primates | Cercopithecidae | *Cercopithecus neglectus* | 99 | 108 | 0.99 | 4.5 | P |
| | | *Macaca fuscata* | 109 | 109 | 1.00 | 16 | P |
| Rodentia | Caviidae | *Dolichotis patagonum* | 99 | 111 | 0.98 | 8 | D |
| | | *Galea musteloides* | 75 | 106 | 0.86 | 0.34 | P |
| | Muridae | *Meriones unguiculatus* | 66 | 95 | 0.88 | 0.06 | P |
| Scandentia | Tupaiidae | *Tupaia glis* | 84 | 107 | 0.92 | 0.18 | P |
| Diprotodontia | Macropodidae | *Macropus fuliginosus* | 96 | 119 | 0.92 | 43.5 | Pe |
| | | *Macropus giganteus* | 101 | 122 | 0.93 | 41 | Pe |

**Note:**
Abbreviations for the locomotor modes: D, digitigrade; P, plantigrade; Pe, pentadactyl; U, unguligrade.

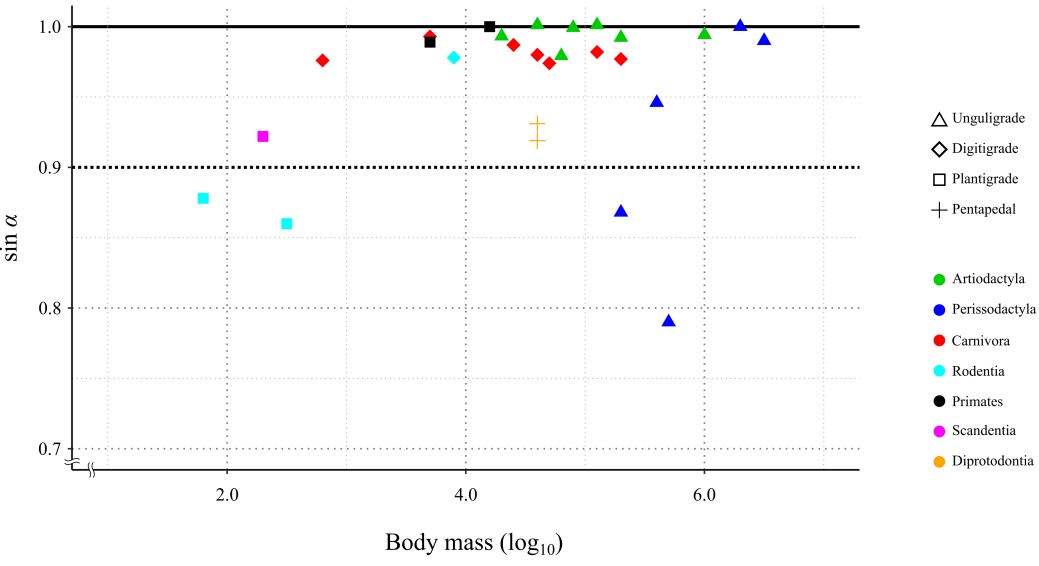

**Figure 6 Relationship between the efficiency of the calcaneus lever (sin $a$) during the stance phase in the studied species.** X-axis shows the body mass in $\log_{10}$. Y-axis shows the value of sin $\alpha$. See legends for the taxonomic group (order) of the target animals (color-code) and the locomotor mode (symbol shape).                         

mechanical advantage of the calcaneal lever among locomotor modes (Table 4). Additional phyANOVA and pairwise comparisons were also conducted among orders. However, since Scandentia was represented by only one species, it was excluded from the analysis. The phyANOVA among orders also showed no significant differences ($p = 0.665$), and all pairwise Holm-adjusted $p$-values were equal to 1.00 (Table 4). Thus, neither locomotor

**Table 4 Results of phyANOVA for the relationships between locomotor modes and the efficiency of the calcaneal lever of the ankle joint extensor (sin $\alpha$), and between taxonomic order and sin $\alpha$.**

**PhyANOVA (Locomotor mode *vs.* sin $\alpha$)**

|  | Sum Sq | Mean Sq | *F* value | *p*-value |
|---|---|---|---|---|
| Locomotor mode | 0.0143 | 0.0048 | 3.4994 | 0.618 |
| Residual | 0.0299 | 0.0014 | — | — |
| Pairwise posthoc test | | | | |
| Comparison | | | Adjust *p*-value | |
| Digitigrade *vs.* Unguligrade | | | 1 | |
| Digitigrade *vs.* Plantigrade | | | 1 | |
| Digitigrade *vs.* Pentapedal | | | 1 | |
| Unguligrade *vs.* Plantigrade | | | 1 | |
| Unguligrade *vs.* Pentapedal | | | 1 | |
| Plantigrade *vs.* Pentapedal | | | 1 | |

**PhyANOVA (Order *vs.* sin $\alpha$)**

|  | Sum Sq | Mean Sq | *F* value | *p*-value |
|---|---|---|---|---|
| Order | 0.0240 | 0.0048 | 5.068 | 0.661 |
| Residual | 0.0180 | 0.0009 | — | — |
| Pairwise posthoc test | | | | |
| Comparison | | | Adjust *p*-value | |
| Artiodactyla *vs.* Perissodactyla | | | 1 | |
| Artiodactyla *vs.* Carnivora | | | 1 | |
| Artiodactyla *vs.* Primates | | | 1 | |
| Artiodactyla *vs.* Rodentia | | | 1 | |
| Artiodactyla *vs.* Diprotodontia | | | 1 | |
| Perissodactyla *vs.* Carnivora | | | 1 | |
| Perissodactyla *vs.* Primates | | | 1 | |
| Perissodactyla *vs.* Rodentia | | | 1 | |
| Perissodactyla *vs.* Diprotodontia | | | 1 | |
| Carnivora *vs.* Primates | | | 1 | |
| Carnivora *vs.* Rodentia | | | 1 | |
| Carnivora *vs.* Diprotodontia | | | 1 | |
| Primates *vs.* Rodentia | | | 1 | |
| Primates *vs.* Diprotodontia | | | 1 | |
| Rodentia *vs.* Diprotodontia | | | 1 | |

mode nor order showed significant differences in the mechanical advantage of the calcaneal lever.

Since the mechanical advantage of the calcaneal lever was not normally distributed, the Spearman's rank order correlation was used to assess the correlation coefficient. The correlation coefficient between the $\theta_{obs}$ and the $\theta_{est}$ was 0.75, with the *p*-value less than 0.05 ($p = 1.21 \times 10^{-5}$: Table 4 and Fig. 7). This indicated a strong positive correlation between $\theta_{obs}$ and $\theta_{est}$, and supporting a significant correlation in the population.
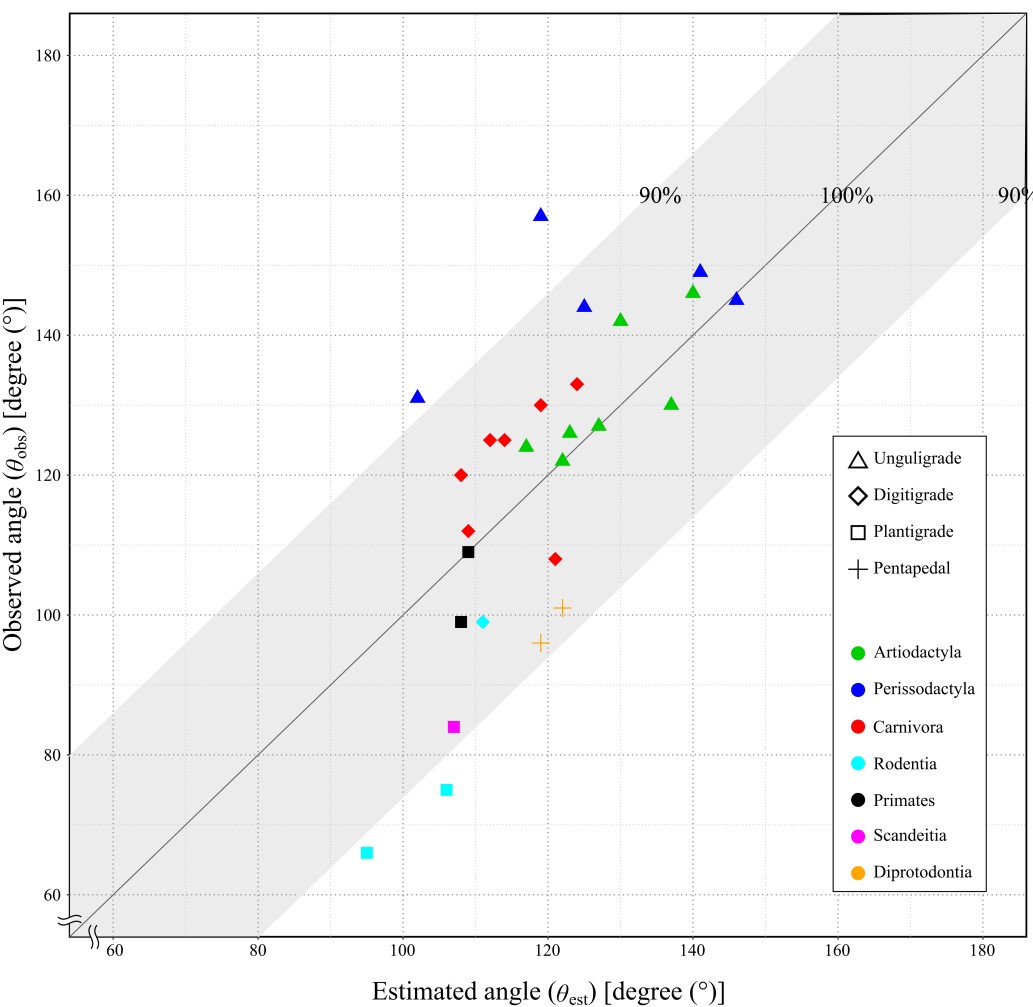

**Figure 7 Relationship between the observed angle ($\theta_{obs}$) and the estimated angle ($\theta_{est}$) in studied species.** X-axis shows the estimated angle ($\theta_{est}$). Y-axis shows the observed angle ($\theta_{obs}$). The gray area shows the lever of the ankle joint exerts more than 90%. See legends for the taxonomic group (order) of the target animals (color-code) and the locomotor mode (symbol shape).

We also performed multiple linear regression analysis *via* PGLS using three data sets: (1) Kiso horse data representing horse; (2) thoroughbred data representing horse; and (3) excluding horse to examine the relationships between $\theta_{obs}$, $\theta_{est}$, and body mass. This approach was taken because the mechanical advantage of the calcaneal lever values differed notably between the Kiso horse and the thoroughbred, even though they belong to same species (Table 3). The regression equation for the $\theta_{obs}$ can be expressed as:

$$\theta_{obs} = \beta_0 + \beta_1 \cdot \theta_{est} + \beta_2 \cdot \text{body mass} \tag{3}$$

In the Kiso horse data set, the regression coefficients were $\beta_0 = 4.90$ (95% confidence interval (CI) [−34.25 to 43.05], intercept), $\beta_1 = 0.48$ (95% CI [0.12–0.83], coefficient for $\theta_{est}$, $p = 0.009$), and $\beta_2 = 4.87$ (95% CI [3.10–6.25], coefficient for body mass, $p = 2.9 \times 10^{-6}$), with a significant overall model ($p = 6.1 \times 10^{-8}$, adjusted $R^2 = 0.74$) (Table 5). In the

**Table 5 Results of PGLS of observed ankle joint angle ($\theta_{obs}$) as a function of estimated ankle joint angle ($\theta_{est}$) and body mass (ln).**

PGLS of $\theta_{obs}$ as a function of $\theta_{est}$ and body mass (ln)

| Dataset | Predictor | Estimate | Std. Error | t value | p value | Pagel's $\lambda$ |
|---|---|---|---|---|---|---|
| Kiso horse | Intercept | 4.90 | 18.94 | 0.26 | 0.798 | 0.943 |
| | $\theta_{est}$ | 0.48 | 0.17 | 2.84 | 0.009 | |
| | Body mass | 4.87 | 0.76 | 6.14 | $2.89 \times 10^{-6}$ | |
| Thoroughbred | Intercept | 8.42 | 21.99 | 0.38 | 0.705 | 0.977 |
| | $\theta_{est}$ | 0.43 | 0.19 | 2.28 | 0.032 | |
| | Body mass | 4.87 | 0.86 | 5.68 | $8.7 \times 10^{-6}$ | |
| Excluding horse | Intercept | 5.46 | 18.75 | 0.29 | 0.773 | 0.938 |
| | $\theta_{est}$ | 0.47 | 0.17 | 2.85 | 0.009 | |
| | Body mass | 4.61 | 0.76 | 6.08 | $4.0 \times 10^{-6}$ | |

Thoroughbred data set, the coefficients were $\beta_0 = 8.42$ (95% CI [−37.85 to 54.69]), $\beta_1 = 0.43$ (95% CI [0.03–0.83], $p = 0.032$), and $\beta_2 = 4.87$ (95% CI [3.07–6.67], $p = 8.7 \times 10^{-6}$) (overall model $p = 6.7 \times 10^{-7}$, adjusted $R^2 = 0.68$) (Table 5). In the data set excluding *Equus caballus*, the coefficients were $\beta_0 = 5.46$ (95% CI [−33.30 to 44.22]), $\beta_1 = 0.47$ (95% CI [0.12–0.82], $p = 0.009$), and $\beta_2 = 4.61$ (95% CI [2.97–6.25], $p = 4.0 \times 10^{-6}$) (overall model $p = 9.5 \times 10^{-8}$, adjusted $R^2 = 0.75$) (Table 5). Pagel's $\lambda$ values indicated a strong phylogenetic signal across all data sets (0.943, 0.977, and 0.938, respectively) (Table 5).

## DISCUSSION

A total of 85.2% (23 out of 27) of the studied species showed a mechanical advantage of the calcaneal lever greater than 0.9 (sin $\alpha > 0.9$). This species included individuals with a wide range of body masses, from 0.18 kg (common tree shrew) to 3,400 kg (white rhinoceros), as well as species representing three locomotor modes and six orders (Table 3). The phyANOVA and pairwise comparison tests for mechanical advantage of the calcaneal lever across locomotor modes and taxonomic orders detected no significant differences. Additionally, the correlation coefficient between mechanical advantage of the calcaneal lever and body mass was not statistically significant ($p = 0.12$), suggesting that the mechanical advantage of the calcaneal lever is not strongly influenced by these factors. These results suggest that FAEA of terrestrial mammals is maintained within a range that optimizes mechanical advantage of the calcaneal lever during the stance phase under steady-speed walking conditions, across a wide range of body masses and taxa. A higher mechanical advantage of the calcaneal lever enhances torque generation at the ankle joint for a given muscle force, contributing to more efficient support and propulsion during walking (*Biewener, 1989*; *Biewener & Roberts, 2000*; *Ray & Takahashi, 2020*). This mechanical advantage is particularly important during the stance phase, when joints must resist gravitational flexion forces.

Based on the multiple linear regression analyses using PGLS, the Kiso horse data set more closely resembled the excluding *E. caballus* data set (Table 5); therefore, the Kiso

**Table 6 The ln of body mass (BM), mechanical advantage of the calcaneal lever (sin $\alpha$), observed ($\theta_{obs}$) and estimated ($\theta_{est}$) ankle joint angles, the difference between $\theta_{obs}$ and $\theta_{est}$ ($\theta[sub]$).**

| Species | ln BM | sin $\alpha$ (mechanical advantage) | $\theta_{obs}$ | $\theta_{est}$ | $\theta_{obs}-\theta_{est}$ | $\theta_{r-obs}$ | $\theta_{obs}-\theta_{r-obs}$ |
|---|---|---|---|---|---|---|---|
| *Meriones unguiculatus* | 4.1 | 0.87 | 66 | 95 | −29 | 69 | −3 |
| *Tupaia glis* | 5.2 | 0.92 | 84 | 107 | −23 | 80 | 4 |
| *Galea musteloides* | 5.8 | 0.86 | 75 | 106 | −31 | 83 | −8 |
| *Suricata suricatta* | 6.6 | 0.99 | 108 | 121 | −13 | 93 | 15 |
| *Cercopithecus neglectus* | 8.4 | 0.99 | 99 | 108 | −9 | 96 | 3 |
| *Felis silvestris catus* | 8.5 | 1.00 | 112 | 109 | 3 | 96 | 16 |
| *Dolichotis patagonum* | 9.0 | 0.98 | 99 | 111 | −12 | 100 | −1 |
| *Macaca fuscata* | 9.7 | 1.00 | 109 | 111 | −2 | 103 | 6 |
| *Capra hircus* | 9.9 | 0.99 | 130 | 137 | −7 | 116 | 14 |
| *Chrysocyon brachyurus* | 10.1 | 0.99 | 133 | 124 | 9 | 111 | 22 |
| *Canis lupus* | 10.5 | 0.98 | 130 | 119 | 11 | 111 | 19 |
| *Capricornis crispus* | 10.6 | 1.00 | 127 | 127 | 0 | 115 | 12 |
| *Macropus fuliginosus* | 10.6 | 0.92 | 96 | 119 | −23 | 111 | −15 |
| *Macropus giganteus* | 10.7 | 0.93 | 101. | 122 | −21 | 113 | −12 |
| *Acinonyx jubatus* | 10.8 | 0.97 | 125 | 112 | 13 | 109 | 16 |
| *Ammotragus lervia* | 11.1 | 0.98 | 142 | 130 | 12 | 118 | 24 |
| *Cervus nippon yesoensis* | 11.2 | 1.00 | 126 | 123 | 3 | 116 | 10 |
| *Panthera tigris* | 11.7 | 0.98 | 125 | 114 | 11 | 114 | 11 |
| *Rangifer tarandus* | 11.7 | 1.00 | 121 | 122 | −1 | 118 | 3 |
| *Choeropsis liberiensis* | 12.1 | 0.99 | 124 | 117 | 7 | 117 | 7 |
| *Panthera leo* | 12.2 | 0.98 | 120 | 108 | 12 | 113 | 7 |
| *Tapirus terrestris* | 12.3 | 0.88 | 131 | 103 | 28 | 111 | 20 |
| *Equus caballus: Kiso horse* | 12.9 | 0.95 | 144 | 125 | 19 | 125 | 19 |
| *Equus caballus: thoroughbred* | 13.2 | 0.79 | 157 | 119 | 38 | 123 | 34 |
| *Giraffa reticulata* | 13.8 | 0.99 | 146 | 140 | 6 | 136 | 10 |
| *Rhinoceros unicornis* | 14.6 | 1.00 | 145 | 146 | −1 | 142 | 3 |
| *Ceratotherium simum* | 15.0 | 0.99 | 149 | 141 | 8 | 142 | 7 |

horse data set was treated as representative of *E. caballus*. The regression equation showed that the effect of body mass on $\theta_{obs}$ is notably stronger than that of $\theta_{est}$, where each unit increase in $\theta_{est}$ leads to an increase in $\theta_{obs}$ by about 0.48 degrees. Both factors significantly influence $\theta_{obs}$, with body mass having a more substantial impact. However, in species exceeding 10 kg in body mass, the recalculated $\theta_{obs}$ values using the regression equation (hereafter referred to as $\theta_{r-obs}$) showed larger differences from $\theta_{obs}$ than differences of $\theta_{est}$ values. These mean values were 11.1 and 6.0 degrees, respectively. The $\theta_{r-obs}$ values showed smaller differences than $\theta_{est}$ in 11 out of 26 species, but five of these 11 species had a body mass below 10 kg (Mongolian gerbil, common tree shrew, yellow-toothed cavy, meerkat, and cat) (Table 6). Notably, only seven out of the 26 species had a body mass below 10 kg (Table 6). Among the 19 species exceeding 10 kg, three species (pygmy hippopotamus, tiger, and Kiso horse) showed identical differences from the $\theta_{obs}$ for both $\theta_{r-obs}$ and $\theta_{est}$. In

the remaining 16 species, $\theta_{est}$ showed smaller differences than $\theta_{r-obs}$ in 11 cases. On average differences of $\theta_{r-obs}$ were 3.8 degrees in species below 10 kg, and 11.1 degrees in species exceeding 10 kg. In contrast, the differences in $\theta_{est}$ were −16.3 degrees in species below 10 kg, and 6.0 degrees in species exceeding 10 kg (Table 6). In addition, the correlation coefficient between the $\theta_{obs}$ and the $\theta_{est}$ was significantly positive and strong ($r = 0.74$). Therefore, $\theta_{est}$ can be reliably used to reconstruct $\theta_{obs}$ in species exceeding 10 kg, even if the skeletal specimens have been stored disarticulated.

Notably, kangaroos (*Macropus* spp.) showed negative values for $\theta_{obs}$ minus $\theta_{est}$ (−23 and −21, respectively) (Table 6), indicating that the $\theta_{obs}$ were more flexed than the $\theta_{est}$. Kangaroos employ a unique form of locomotion known as pentapedal, which uses the tail to support body mass during quadrupedal walking (*Windsor & Dagg, 1971*; *O'Connor et al., 2014*). However, because the tail is used during the swing phase of the hindlimbs, other factors may contribute to this result. One possible explanation is that quadrupedal walking is not the typical mode of locomotion for them; they primarily use hopping as their main mode of locomotion. The hopping gait is most commonly employed during regular movement, escape responses, and long-distance travel (*Windsor & Dagg, 1971*; *O'Connor et al., 2014*). Their hindlimbs are highly specialized for hopping, *e.g.*, elongated metatarsals and shortened forelimbs (*Alexander & Vernon, 1975*; *Polly, 2007*; *McGowan & Collins, 2018*; *Jones, Travouillon & Janis, 2024*). The elongated metatarsals of hopping species increase the distance between the ground and the ankle joint, whereas short forelimbs reduce the distance between the ground and the trunk. It is possible that flexion of the hindlimb joints during quadrupedal walking, as observed in this study, serves to lower the posterior trunk, which might partly account for the more flexed $\theta_{obs}$ relative to $\theta_{est}$ in kangaroos. However, because this study includes two kangaroos species, additional data from other hopping species such as wallaby and kangaroo rat are needed to confirm the idea.

The data from two breeds of horses, the Kiso horse and the thoroughbred, implied that artificial selection had influenced on their morphology. These two breeds are classified in the same species, *Equus caballus* (horse), but their sin $\alpha$ differed by 0.16, with the Kiso horse showing higher value (0.95) than the Thoroughbred (0.75) (Table 3). This difference was greater than between the highest and the second lowest sin $\alpha$ of this study, 0.13 (Table 3). This difference appeared to result from more extended hindlimb posture in thoroughbred, where $\theta_{obs}$ exceeded $\theta_{est}$, reducing the mechanical advantage of the calcaneal lever. While both breeds are domesticated, their morphology reflects fundamentally different breeding purposes. The Kiso horse is a medium-sized Japanese native horse that had been raised since around fifth century and used for agriculture and transportation in mountain regions until the middle of the 20th century (*Nakagawa, 2014*; *Yamashita, 2014*). Today, it is maintained primarily for conservation. In addition, the Japanese native horses have remained independent from modern breeding (*Nozawa, 1992*). Thoroughbred, on the other hand, was developed in the late 18th century and has been strongly selected for racing abilities and characteristics until today (*Cunningham, 1991*; *Bower et al., 2012*). It is possible that artificial selection for enhanced galloping ability in Thoroughbreds has favored morphologies that maximize propulsion during high-speed

locomotion. This may include joint configurations that enhance the rotational torque around the ankle joint (*Gnagey, Clayton & Lanovaz, 2006*). Additionally, efficient elastic energy storage requires significant tendon stretch, which is facilitated by joint flexion (*Dimery, Alexander & Ker, 1986*; *Biewener, 1998*). Therefore, more extended posture during walking of thoroughbred may reflect an adaptation for maximize both elastic energy recovery and mechanical advantage of the calcaneal lever during running. However, this study had only two breeds of horses because we aimed to collect data from wide range of taxa. To test this idea, it will be necessary to collect data from broad range of species and breeds within Equidae.

There were some possibilities to lead errors in this study. *In vivo* measurement was limited by animal behavior and shooting environment. In addition, the measurements were not performed with any markers directly put on the living specimens. When measuring skeletal specimens, although the articulation flexibility is small, thickness of the adhesive material could cause errors, and the effect of measurement tools accuracy also coursed errors. The ideas to avoid these errors, such as using motion capture, making a strait walkway for recording video, and taking video with x-ray cinematography. Furthermore, no speed-related parameters, such as actual velocity, Froude number, or duty factor, were measured in this study. Although all animals were observed walking at self-selected speeds, variation in walking velocity among species may have affected joint postures. Therefore, the lack of speed standardization represents a limitation when comparing $\theta_{obs}$ values across taxa. Future studies may benefit from quantifying stride parameters to better control for interspecific variation in locomotor dynamics.

## CONCLUSION

In this study, we proposed a mechanical model of the ankle extensor apparatus of terrestrial mammals that approximates the ankle joint as a simple hinge and the force of the ankle extensors as acting parallel to the tibia. Our model predicts an estimated joint angle ($\theta_{est}$) at which the moment arm of the ankle extensors is maximized. By comparing this predicted angle with observed ankle joint angles ($\theta_{obs}$) during walking, we found that despite large variations in $\theta_{obs}$ among species, mechanical advantage of the calcaneal lever ($\sin a$) remains within a relatively narrow range, supporting the notion that, across taxa, skeletal morphology at the ankle joint has been selected for optimization of mechanical advantage of the calcaneal lever for the ankle plantar flexors. This hypothesis is supported even in species with varying ankle joint postures and ranges of motion. This insight provides a new perspective for reconstructing locomotor postures in extinct terrestrial mammals and enhances understanding of the biomechanical constraints shaping limb function.

## ACKNOWLEDGEMENTS

We thank Naomi Wada (Yamaguchi University) for supplying the camera equipment. We thank Tatsuo Oji (Nagoya University, now Nagoya City Science Museum), Naoki Kohno (National Museum of Nature and Science, University of Tsukuba) and Masakazu

Fukuzumi (University of Tsukuba) for providing helpful advice. We also thank reviewers for helping to improve this article.

### Funding
The authors received no funding for this work.

### Competing Interests
The authors declare that they have no competing interests.

### Author Contributions
- Fumihiro Mizuno conceived and designed the experiments, performed the experiments, analyzed the data, prepared figures and/or tables, authored or reviewed drafts of the article, and approved the final draft.
- Shin-ichi Fujiwara conceived and designed the experiments, authored or reviewed drafts of the article, and approved the final draft.

### Animal Ethics
The following information was supplied relating to ethical approvals (*i.e.*, approving body and any reference numbers):

This study videotaped mammals from visitor-viewing route and no treatment.

We followed the ethical guidelines of National Museum of Nature and Science (Tokyo) for dissection.

### Data Availability
The raw measurements are available in the Supplemental Files.

### Supplemental Information
Supplemental information for this article can be found online at http://dx.doi.org/10.7717/peerj.20056#supplemental-information.

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
