# Peer review of "Estimating ankle joint angle from skeletal geometry: a mechanical model of the calcaneal lever in terrestrial mammals"

_PeerJ, doi:10.7717/peerj.20056_

## Round 0.1 · original submission · Major Revisions

Dear Doctor Mizuno, I ask you to carefully improve the manuscript in accordance with the reviewers' fundamental arguments.

**Language Note:** The review process has identified that the English language must be improved. PeerJ can provide language editing services - please contact us at [email protected] for pricing (be sure to provide your manuscript number and title). Alternatively, you should make your own arrangements to improve the language quality and provide details in your response letter. – PeerJ Staff

Reviewer 1 ·

Basic reporting

This paper focuses on the ankle lever mechanism of land mammals, and discusses the problem of deriving the apparent joint Angle from the perspective of the internal structure of the ankle. The research is innovative and important, but there is still room for improvement in some aspects. Here are my recommendations changes.
Comment 1: The author obtained that the Achilles tendon and tibia are parallel by using museum specimens and dissecting animal corpses, but the actual situation of live animals in the walking stage was not collected. To some extent, this cannot prove the conclusion that observed angle and estimated angle are highly correlated mentioned by the author in the conclusion, because it was obtained by observing live animals in zoos. The respondents are different.

Comment 2: The author simplified the organization of the ankle joint. It is well known that the rotation of the ankle joint in animals is not purely circular, so the author needs to explain this simplification.

Comment 3: When establishing the kinematic model, the author adopted the conclusion that the tibia and gastrocnemius are almost parallel in the conclusion, which seems to have a certain logical sequence problem. At the same time, as a mathematical model related to kinematics, the annotation of various elements in Figure 1 should also be standardized, and other graphs also have non-standard drawing.

Comment 4: In vivo measurement is limited by animal behavior and shooting environment, and the measurement distance of different animals is different, which may introduce errors; When measuring bone specimens, although the effect of reassembling and confirming joint flexibility is small, there may still be errors, and the effect of measurement tool accuracy on the results is not mentioned.

Comment 4: In the example shown in Figure 2, only one ankle was analyzed and the other ankle was not analyzed simultaneously, which may affect the reliability of the conclusion.

Comment 5: Although the samples observed by the author cover multiple species, different body weights and movement patterns, the sample size of some taxa is too small, and most of the samples are located in the artificially created living environment of zoos, which further limits the universality of the conclusions of this paper.

Comment 6: The author should set up certain control experiments, such as ankle Angle changes of the same species under different living environments, or compare animal data of different ages and genders to enhance the persuasive conclusion.

Comment 7: The overall presentation needs to be improved to ensure that your content is clearly understood by the reader.

Experimental design

no comment

Validity of the findings

no comment

·

Basic reporting

The idea behind the study is interesting, novel and important to functional morphology and biomechanics, including palaeontological applications. In general it is well executed but it needs some better wording for clarity, some further literature review/citation, and some corrections to the science, as follows.
Sufficient data are shared.


Background:
First 3 sentences don’t make sense- I think I understand “apparent” ankle joint angle (the roughly measurable angle) but what is the “structural” angle? (there is also “observed” and “estimated” angle mentioned in this section [and below], implying 4 distinct things--- please simplify wording throughout to be clearer) The logic that animals “would” maintain this angle isn’t intact. In theory, if moment generating capacity is a key objective, then yes this would be expected. But this objective is not certain; it is controversial. At least change this to “might” or similar. See these references for some examples and more references:
Hutchinson, J. R., Rankin, J. W., Rubenson, J., Rosenbluth, K. H., Siston, R. A., & Delp, S. L. (2015). Musculoskeletal modelling of an ostrich (Struthio camelus) pelvic limb: influence of limb orientation on muscular capacity during locomotion. PeerJ, 3, e1001.
Cox, S. M., Easton, K. L., Lear, M. C., Marsh, R. L., Delp, S. L., & Rubenson, J. (2019). The interaction of compliance and activation on the force-length operating range and force generating capacity of skeletal muscle: a computational study using a guinea fowl musculoskeletal model. Integrative Organismal Biology, 1(1), obz022.
Labonte, D. (2023). A theory of physiological similarity in muscle-driven motion. Proceedings of the National Academy of Sciences, 120(24), e2221217120.  this reference is very important as it explains why small vs. large animals face different constraints, namely maximizing muscle velocity (and gearing) vs. muscle work (and gearing). So this would impact this study’s findings—we’d not expect gastrocnemius function to be maintained, necessarily, with size.

Some further proofreading is needed throughout the MS. Some phrasing is awkward, e.g. Methods summary- “We built a mechanical model of the ankle joint extension.” Probably best phrased as something like “of the ankle joint extensor apparatus”.

Same section- “highest efficiency” this is a problematic word as it means energy out/energy in. Energy is not being quantified in this study. What is meant that the moment arm of the gastrocnemius is maximal at 90 degrees; that is geometrically certain. Other points such as efficiency are higher-level inferences that need more data – e.g., muscle fibre lengths, metabolic energy consumption, etc. Please replace “lever efficiency” throughout as “estimated/maximal moment arm ratio.”

“been videotaped at 420 fps”—this can be omitted as it’s excessive detail for this section. Just “been recorded” is fine.

“In addition, the senior author”- this wouldn’t normally need to be stated. Just “In addition, we”. [same at line 195- “The senior author dissected cadavers”]

Results summary- “tibia even different species.” Incomplete sentence – “even in” is meant? “the transitions of the ankle joint” – “the rotations of the ankle joint” is much clearer. “the[theta]obs” needs a space. “, using the multiple comparison test” is not needed here and not clear enough anyway. “the correlation efficiency between [theta]obs and [theta]est showed around” is unclear wording.

Conclusion- “is high independent” change to “is highly independent”. “are highly related.” this is a big claim [and should be correlated, not related]. With an 0.7 correlation, there is ~30% of cases that violate the optimal angle relationship. The study should be more cautious in reporting the strength of its findings. [also line 234] How can it be justified that a 0.7 is “strong”?

Main MS—as noted above, much more proofreading is needed. It is beyond the scope of this review to correct the text as done above, but those corrections provide examples. In its current form, the MS takes quite a bit of effort to read through and understand. A heavy rewrite is needed to maximize clarify.

Line 51-52 “inverted pendulum” discussion—this is during walking, and since the old studies concluding an “inverted pendulum” mechanism, it has since been recognised that it is a “spring-loaded inverted pendulum”—main reference = Geyer, H., Seyfarth, A., & Blickhan, R. (2006). Compliant leg behaviour explains basic dynamics of walking and running. Proceedings of the Royal Society B: Biological Sciences, 273(1603), 2861-2867. Lines 55-56 the COM is not at the pelvic girdle per se. It is close to it in humans, but tends to be far cranial to it in most (quadrupedal) mammals. In such models, the COM and the hip are assumed to coincide but this applies to humans.

“knee joint angles among mammalian species are constant during the stance phase”- I’m not so convinced the literature shows this. E.g.
~25 degree flexion in Ren, L., Butler, M., Miller, C., Paxton, H., Schwerda, D., Fischer, M. S., & Hutchinson, J. R. (2008). The movements of limb segments and joints during locomotion in African and Asian elephants. Journal of Experimental Biology, 211(17), 2735-2751. [it’s a shame that the study did not include elephants, for which plenty of data exist and would expand the size + locomotor mode range]
Similar in diverse mammals—
Pike, A. V. L., & Alexander, R. M. (2002). The relationship between limb-segment proportions and joint kinematics for the hind limbs of quadrupedal mammals. Journal of Zoology, 258(4), 427-433.
And so on. This section needs some rethinking.

Line 84- “works” avoid this term as in mechanics it means force x distance; use “functions”.

Line 93 – “an effective angle” again, new implied terms are being introduced—please stick to minimal jargon. Here, it seems “an optimal angle” is meant, but some care should be used in choosing consistent terms for the various angles used in the study.

Line 101 does it matter that other muscles such as digital flexors also cross the ankle joint? They are not mentioned.

Line 114 “The sin [alpha] 90%, where the angle [alpha] ranges from 64° to 116°.”—this logic lost me.

Methods section- this is mostly clear enough. In the in vivo data collection section, mention the number of steps used (they’re in table 1 but this is important in the text too).

“Inkscape (Inkscape project)”—give website

Speed, Froude number or duty factor or similar stride parameters were not calculated to ensure some sort of comparable speed- can these be provided? It seems that only walking was studied, yes? (at animal-chosen speed)

Line 171- what camera was used? Same as in the in vivo analyses?

Line 173 “Two lines were then drawn with the following steps on each picture”—In Inkscape too? Please explain further.

Line 187- what is “other” locomotor mode?
Phylogenetic comparative methods were not used here and arguably they should be. Please add them or provide strong justification why not.

Line 195- “dissected cadavers from zoos to the NMNS” is it meant that the cadavers were donated from zoos to the NMNS? Were all individuals adults? Please specify in in vivo, skeletal and dissection specimens?

The order of the Results (and Discussion, where necessary) should follow the order of the Methods, to make things easier to follow.

Lines 210 and 213 have swapped the Indian and White rhino. Please fix.

Lines 250-251 can an equation, with 95% confidence intervals or similar, be provided here, in case others want to use it? It should be made clear under what conditions this method can be used; it’s restricted to the scope of the data used, so presumably for steady-speed walking, and so on.

Line 266- “heavier body masses had extending”: this is important and needs better phrasing. Greater [theta]obs than [theta]est is what is meant.

Line 275- this paragraph needs clearer wording. It’s, in brief, arguing that bigger species use more optimal ankle angles during walking but not running, because running is not so important to their survival. It gets confusing here as “large herbivores” are mentioned on line 286 but “relatively small” on line 282. And it’s not made clear what, in terms of optimal ankle angle, smaller mammals are “using” it for.

Line 285- “In addition, the joint angles are flexed during the landing of rapid locomotion (Muybridge,1957).” Early stance phase is meant here, and Muybridge did not provide such data on flexion. Just images (and not truly scientific; more artistic). Please cite scientific sources, many of which are cited in the Introduction.

Line 289 “even small irregularities such as pebbles, twigs, and bumps” etc.—I’m sorry, but the ad hoc logic here escapes me. Please flesh out the logic more. The study’s strength is in its mathematical modelling. But substrate roughness is not part of that modelling, so how can this claim be made? Same for the kangaroo; there are no calculations made for pentapedal locomotion, so the logic isn’t thorough.

Section beginning at line 300: this is really interesting but again the logic isn’t made explicit and watertight (e.g. mathematical)—why is “the low sin [alpha] of Thoroughbred would be caused by high adaptation for galloping.”?

I also found that the Conclusion would be hard for some readers to follow. Please make the logic very clear.

Figures and Tables:
Figure 1’s text is hard to read in some parts; it gets crowded, and the colours fade into the background. Given its importance, please reconsider the design.

Figure 4: did it matter how the ankle angle was posed in each specimen, and how that angle related to in vivo angles? (was it sufficiently representative for that individual/taxon?)

Figure 5: x-axis should be Frame number; y-axis should not say “The”.

Figure 6 caption uses “a” not “[alpha]”.

Figure 8 caption mentions a grey area but none is shown.
Hyracoidea is noted but not in Table 1 etc?

Table 2- Proboscidea was studied here but not elsewhere in the paper. This isn’t mentioned or justified; please do.

Table 3- is pentapedal the “other” locomotor mode elsewhere? See comment above.

Table 4 does not need to be a Table and can just be in the main text. Maybe Table 8, too. Also Table 9.

Experimental design

See my comments above- e.g., reconsider phylogenetic comparative methods.

The study falls within PeerJ's scope. The question is well phrased and investigated except where noted above.

Validity of the findings

A main concern is the potential overinterpretation of the correlations, as noted above.

---

## Round 0.2 · Major Revisions

Dear Dr. Mizuno, I ask you to carefully study the reviewers' comments and make serious corrections to the manuscript. It is necessary to provide reasoned responses to the reviewers on each of the points indicated in the review (except for those that you corrected and with which you agreed). I hope that the new version of this article will be approved for publication.

**PeerJ Staff Note**: Please ensure that all review, editorial, and staff comments are addressed in a response letter and that any edits or clarifications mentioned in the letter are also inserted into the revised manuscript where appropriate.

**PeerJ Staff Note**: It is PeerJ policy that additional references suggested during the peer-review process should only be included if the authors agree that they are relevant and useful.

**Language Note**: The review process has identified that the English language must be improved. PeerJ can provide language editing services - please contact us at [email protected] for pricing (be sure to provide your manuscript number and title). Alternatively, you should make your own arrangements to improve the language quality and provide details in your response letter. – PeerJ Staff

Reviewer 1 ·

Basic reporting

no comment

Experimental design

no comment

Validity of the findings

no comment

Additional comments

No comment

·

Basic reporting

Please conduct another round of proofreading; I noticed numerous new errors in the text that was revised. e.g. "This model helps reconst postures in extinct species.” has an error

I thank the authors for their attentive revisions. They've improved the MS substantially.

"The ankle joint angle, typically measured between the tibia and metatarsus, tends to remain relatively constant within species but varies across taxa. "-- this implies that the ankle never moves? Reword to make the meaning + context clearer.

My prior critique "Speed, Froude number or duty factor or similar stride parameters were not calculated to ensure some sort of comparable speed- can these be provided? It seems that only walking was studied, yes? (at animal-chosen speed)" could be accomodated by acknowledging this important limitation in the Discussion.

Otherwise, the paper is in better shape and I appreciate the efforts made here. The change to use PCM will make this paper better received by the community.

Experimental design

no further comments

Validity of the findings

no further comments

Additional comments

no further comments

Reviewer 3 ·

Basic reporting

Article is reasonably well-written, especially considering potential language barriers to English from a Japanese lab. Literature was well-cited but needed multiple references from the biomechanics field to strengthen arguments or improve validity and clarity, which I included in the review document. Standard format was followed, figures had errors but did not appear to have manipulated or fabricated data. Raw data was provided and clearly laid out. Article is self-contained.

Experimental design

The primary research question was original, though its definition within the manuscript could be improved; I made comments accordingly in the review document. The methods clearly address the question rigorously and to ethical standards and were communicated very clearly, easily reproducible.

Validity of the findings

Benefit to literature seemed clear to me but was not well-stated within the article itself; I made comments accordingly in the review document. Data and statistics were rigorous and sound. Conclusions were mostly sound, though I left comments to improve.

Additional comments

General Comments
Replace all mentions of “moment arm ratio” with “mechanical advantage” or “effective mechanical advantage”. This term is common in mechanics literature and correctly signals what you’re referring to as internal/external moment arm ratio, which can be confused with its inverse, gear ratio, which is the external/internal moment arm ratio.
For all sentences in the discussion, please add a topic sentence that tells the reader where you’re going. I find that they have interesting information, but the purpose behind the paragraph is always unclear.
Specific Comments
Lines 14-15/Introduction: This is a very small point, but the abstract states that hindlimbs play a crucial role in enabling efficient locomotion. This is true, obviously, but that idea isn’t really at the core of your paper and isn’t discussed any further. Instead of a major edit, consider removing this comment from the abstract, as it sets my expectations up for something that isn’t featured consentently in the manuscript.
Lines 45-46: I understand the spirit of this sentence but it reads as if you KNOW you’ve correctly reconstructed the joint posture, which isn’t possible as there are no direct observations. Please change to reflect that you’re performing an educated estimate; could consider “This model informs postural reconstructions in extinct species”.
Line 54: Please add “optimized” or “optimal” before the word “stiffness” to reflect that stiffness isn’t just high (ie hard to move) but appropriate to allow deformation and strain energy storage/release.
Lines 80-82: This sentence’s wording needs to be improved, as it currently reads as if bones create force. Of course, the force is created through actuation by ankle plantar flexors; I think it would be better to combine this idea into the existing sentence found lines 82-84 to represent the bones’ motion as a result of muscular forces. This also preserves your transition into the following sentences.
Line 96: The editing process seems to have left too many spaces between sentences.
Line 98: This sentence reads like the calcaneal tuber produces force, which it cannot. You could replace “produce” with “propagation”, or just refer to the calcaneal tuber as the lever through which plantar flexor force is applied.
Lines 98-100: I would strongly recommend referring to the moment arm as the “internal moment arm” and adding “muscle” before “force on line 100. This is the common term used in the biomechanical literature and avoids potential confusion with the external moment arm. This is particularly helpful when talking about greater mechanical advantage, which is really the point you’re making: greater relative internal moment arms make it easier to produce a given force. I would also consider a citation at the end of line 100 as these concepts aren’t something previously introduced.
Lines 108: Unless this format is some specific requirement of the journal, I think you should use this paragraph to restate i) the use of particular angle (what is it called? I would name it here!) and how it physiologically and mechanically relevant, and ii) restate your purpose. What insight does calculating this angle give us on the back end, and how will it help us understand other species structure in the future? This paragraph really falls flat in its current form.
Line 138: Figure 1 does not include an θest term, though it is mentioned in the text. Figure 1’s caption states that blue fans are “the angle (θ)”, and yet they are shown and described in the text as α. Please clarify accordingly in the text and figure.
Lines 118: torque is abbreviated again, no need to abbreviate again the next two times.
Line 125: C is not defined in the text, only in the figure; please add.
Line 132: You’ve already defined ankle angle as θ, no need to do again.
Figure 2: Remove “the” from the title of figure 2.
Lines 203-211: There are multiple abbreviations here that have already been defined, edit accordingly to include only the first instance.
Lines 230-234: Could you provide more details about what positions you tested to ensure that this was the case? I want to be clear that I agree with you conclusion, but Figure 4 seems to show multiple species, which is good, but at a range of different joint angles. I think additional text is needed to assure the reader that parallel orientation of tibia and achilles was present at ALL joint positions common to the different phases of gait stance across ALL species. And, if any exceptions exist, just note them.
Line 281: I would suggest removing “according to the results” and simply state the finding.
Line 288: Again, I believe that the angle between line of action and calcaneal tuber should just be named by you in this paper. Establishing it is the purpose of the manuscript, you should reflect that here and even in your title. Perhaps Achilles moment arm angle?
Lines 289-291: Please replace “muscle function” with “mechanical advantage”. It isn’t a good idea to speculate the optimization of ankle plantar flexor function without specific muscular data from gait, which varies wildly between species particularly during specific gait periods. You should also include, either here or in the introduction, some references to why increased mechanical advantage is beneficial or not to gait efficiency (ie but only among human literature of which I’m familiar: https://pmc.ncbi.nlm.nih.gov/articles/PMC11429282, https://www.nature.com/articles/s41598-020-65626-5 .)
Lines 291-295: I don’t think you should focus on extension as the primary feature of walking enabled by increased effective mechanical advantage at the ankle. Instead, you should talk about propulsion (ie the amount of positive work or energy generated). As your data suggests, if mechanical advantage is relatively constant throughout stance across species, then your angle is evidence of a feature contributing towards enhancing propulsion at a reduced metabolic cost per muscle of investigation (here you just looked at the ankle). (ie https://pubmed.ncbi.nlm.nih.gov/12673146/ , https://journals.physiology.org/doi/full/10.1152/japplphysiol.00003.2004?utm_source=chatgpt.com .)
Lines 296-323: Needs a topic sentence. This reads like a results section with very little interpretation. Please include examples of animals where this angle could be draw some meaningful conclusions.
Lines 325-332: This conclusion is not accurate, as muscular activation can be increased to increase the linear force component of torque. The last sentence also directly counteracts multiple statements you made in your introduction. Consider removing or editing for logical consistency.
Lines 334-352: As you state, hopping gait is the dominant locomotive mode of kangaroos, thus their kinematics differing isn’t surprising. But, I believe you should stress that increased ankle flexion is most likely a feature of increasing energy storage through joint strain (ie ankle angle change), and thus increasing strain in the Achilles and the plantaris, whose tendon crosses all distal foot joints. Thus, I would consider this an important caveat of using this angle to reconstruct extinct species’ of species relying on large joint angle changes, as it would not describe the mechanical advantage relationship you’ve displayed. You do mention this is relation to horse gaits, but here, kangaroos are much more extreme users of elastic strain energy (https://onlinelibrary.wiley.com/doi/10.1111/joa.12715 , https://journals.biologists.com/jeb/article/198/9/1829/7091/In-Vivo-Muscle-Force-and-Elastic-Energy-Storage ). You may consider a similar analysis in hopping species going beyond wallaby and kangaroos, or even in bipedal species in the future.

Lines 346: This isn’t a complete sentence; not sure what’s trying to be communicated or I would have a suggestion.
Line 348: Give the directionality of the difference in moment arm ratio, as in which has a higher or lower ratio so that the reader doesn’t have to refer to a table.
Line 349: This sentence may have been edited too much. It reads better as “While both breeds are domesticated, their morphology reflects fundamentally different purposes in their breeding. Kiso horses are a medium-sized, native Japanese horse bred since the fifth century for agriculture and transportation…”
Lines 346-369: The logic in this paragraph is not clear. You begin with why these two horses likely differ in moment arm ratios due to their functions (Kiso looks to be more general, thoroughbred more specific), but then describe features of all horse anatomy. In line 368, you state you will need more species to “confirm this idea”, but it is not clear what that idea is! I would recommend stating it explicitly in the first sentence. Also, as the comment above states, it isn’t clear what species has a greater moment arm ratio, and therefore this paragraph is lacking a discussion of how exactly greater mechanical advantage is beneficial for running performance and but wouldn’t also be advantageous for locomotion over rough terrain (ie farmland, mountainous or rocky paths).
Lines 381-387: I’m not confident these sentences make sense based on my concerns laid out in previous comments. Lower body mass of course reduces required torques because the mass you’re moving is smaller, but the relative torque contributions are not necessarily lower (https://journals.plos.org/plosone/article?id=10.1371/journal.pone.0078392 ). Posture is much more influential than mass in determining ankle angle motion throughout stance (https://www.science.org/doi/abs/10.1126/science.2740914 ), and thereby moment arm ratios, particularly in relation to the actual flexion values (ie rodentia in figure 3 have much more extended ankles). This sentence should account for this nuance, and I am not seeing a convincing argument for lesser body mass relating consistently with reduced torques when normalized to body mass. Though this idea extends outside of my literature knowledge, I think it might be wise to highlight smaller animals likely having more crouched gait to make escape from predators more easily, both in faster acceleration and ability to hid, as well as their own food sources being easier access. I believe these factors would be much better descriptors of moment arm ratios than body mass.
Lines 395-396: Some small tweaks to this clause would really increase its impact. Consider the following: “…supporting the notion that, across taxa, skeletal morphology at the ankle joint reflects has been selected for optimization of mechanical advantage for the ankle plantar flexors.”
Lines 397-398: Consider revising to the following or something similar: “This hypothesis is supported even in species with varying ankle joint postures and ranges of motion”.

---

## Round 0.3 · Minor Revisions

Dear Dr. Mizuno, I ask you to carefully review the terminology in this article in accordance with the reviewer's comments.

·

Basic reporting

I generally am satisfied enough by the changes, and see that Reviewer 3 has prompted some very good changes too. But one new concern has arisen in revision: again, I disagree with use of "efficiency" in reference to moment arms or lever arms; namely, "efficiency of the calcaneal lever". Efficiency should be used to refer to energy output divided by energy input. Moment/lever arms (as used here) do not involve efficiency so that should not be used in reference to their values. Please choose an alternative term. Something like "mechanical advantage" works best, in my opinion, vs. "efficiency".

Experimental design

N/A

Validity of the findings

N/A

Additional comments

N/A

Reviewer 3 ·

Basic reporting

Basic reporting is satisfactory.

Experimental design

Experimental design is satisfactory.

Validity of the findings

Validity of findings is satisfactory.

Additional comments

The authors have revised the manuscript to my satisfaction. I applaud their efforts and appreciate the opportunity to review and contribute to interesting scientific work!

---

## Round 0.4 · accepted · Accept

Dear Dr. Mizuno and Dr. Fujiwara, this article is recommended for publication. I hope that you will continue to conduct such interesting research in the future.